# Impact of COVID-19 on Quality of Life in Long-Term Advanced Rectal Cancer Survivors

**DOI:** 10.3390/healthcare11141981

**Published:** 2023-07-08

**Authors:** Daniel Blasko, Claudia Schweizer, Tim Fitz, Christoph Schröter, Christopher Sörgel, Annett Kallies, Rainer Fietkau, Luitpold Valentin Distel

**Affiliations:** 1Department of Radiation Oncology, Universitätsklinikum Erlangen, Friedrich-Alexander-Universität Erlangen-Nürnberg (FAU), 91054 Erlangen, Germany; dblasko922@gmail.com (D.B.); claudia.schweizer@uk-erlangen.de (C.S.); tim.fitz994@gmail.com (T.F.); schroter.christoph@gmail.com (C.S.); chris-soergel@web.de (C.S.); annett.kallies@uk-erlangen.de (A.K.); rainer.fietkau@uk-erlangen.de (R.F.); 2Comprehensive Cancer Center Erlangen-EMN (CCC ER-EMN), Universitätsklinikum Erlangen, Friedrich-Alexander-Universität Erlangen-Nürnberg (FAU), 91054 Erlangen, Germany

**Keywords:** quality of life, COVID-19, pandemic, rectal cancer, radiochemotherapy

## Abstract

Colorectal cancer remains one of the most commonly diagnosed cancers. Advanced rectal cancer patients receive neoadjuvant radiochemotherapy as well as surgery and suffer from reduced health-related quality of life due to various side effects. We were interested in the role of the COVID-19 pandemic and how it affected those patients’ quality of life. A total of 489 advanced rectal cancer patients from the University Hospital Erlangen in Germany were surveyed between May 2010 and March 2022 and asked to fill out the EORTC QLQ-C30 and QLQ-CR38 questionnaires over eight different time points: at the beginning, during and after radiochemotherapy, right before surgery, and in yearly intervals after surgery for up to four years. Answers were converted to scores to compare the COVID-19 period to the time before March 2020, focusing on the follow-ups, the developments over time—including by sex and age—and the influence of the TNM cT-stage. Overall, a trend of impaired functional and symptom scores was found across all surveys with few significances (body image −10.6 percentage points (pp) after one year; defecation problems +13.5 pp, insomnia +10.2 pp and weight loss +9.8 pp after three years; defecation problems +11.3 pp after four years). cT4-stage patients lost significantly more weight than their cT1-3-stage counterparts (+10.7 to 13.7 pp). Further studies should be conducted to find possible causes and develop countermeasures for future major infectious diseases.

## 1. Introduction

Colorectal cancer is the third most commonly diagnosed cancer, with an estimation of 1.93 million new cases in 2020, and the second leading cause of cancer deaths worldwide with approximately 930,000 deaths in the same year [1], making it a major health concern. Neoadjuvant radio(chemo)therapy followed by surgery is a key element in treating advanced rectum cancer [2] and leads to impairment of patients’ quality of life (QOL) during treatment [3,4], e.g., defecation and urinary problems. Even years after diagnosis, cancer patients’ QOL is significantly impacted [5,6].

The European Organization for Research and Treatment of Cancer (EORTC) has created a reliable and valid cancer specific multidimensional questionnaire, the QLQ-C30 [7], and a colorectal cancer specific module, the QLQ-CR38 [8], to evaluate their QOL and compare it with other populations. As higher QOL is associated with lower mortality in general [9], it plays a crucial role in terms of disability and quality-adjusted life years. Therefore, potentially influencing QOL parameters, such as cancer classification, should be identified and analysed to understand their effects on patient outcomes.

In March 2020, the World Health Organization (WHO) declared the COVID-19 outbreak as a pandemic [10]. As a result, multiple public health measures were implemented in Germany to halt the spread of the disease, varying greatly over time. Surgeries were postponed and relatives were forbidden to visit patients. The constant threat of getting infected affected people’s day-to-day life, leading to higher mental health impairments [11]; the impacts could be seen in cancer patients as well [12]. The aim of this study was to assess the patients’ QOL during treatment and up to four years after surgery, both prior to and after the beginning of the COVID-19 pandemic, and to compare the results. It is especially focusing on the time after surgery and the development throughout all stages, and can be seen as a follow-up to a similar study conducted with an overlapping patient sample for the time before surgery [13].

## 2. Materials and Methods

This study includes 489 patients from the University Hospital Erlangen, Germany, with advanced rectal cancer who were treated with neoadjuvant radiochemotherapy followed by surgical excision of the entire mesorectum, corresponding to a total mesorectal excision (TME). Written informed consent was obtained from all patients at the “front door”, allowing for their participation in the study and the collection of their clinical data. The data were collected between May 2010 and March 2022, with March 2020 to March 2022 being the COVID-19 time. The questionnaires QLQ-C30 and QLQ-CR38 by the European Organization for Research Treatment of Cancer (EORTC) were used for the survey. Both questionnaires have been validated by the EORTC. They consist of 68 questions in total, with item response options ranging from (1) (“not at all”) to (4) (“very much”) for all items except two ((1) (“very poor”) to (7) (“excellent”)). Forty-nine of these questions can be answered by all patients, meanwhile the nineteen other questions refer to different subgroups (males/females and patients with/without a stoma). All the answers were converted to 27 different scores, with each score containing one to seven items. Out of those 27 scores, there are 17 functional scores and 10 symptom scores, both of which were recalculated as percentages (0–100%). For functional scores a higher percentage score is more desirable, whereas for symptom scores a lower percentage score represents a better QOL. As some questions can only be answered by a subgroup and/or include private sexual questions, thus resulting in not enough data, only 23 of the scores have been used in this study. Patients were asked to fill out the questionnaires at the beginning (day 0; questionnaire 1), during (day 14; week 2; questionnaire 2) and at the end (day 35; week 5; questionnaire 3) of radiotherapy, and right before surgery (day 70; week 10; questionnaire 4). Additionally, four annual follow-ups were surveyed: one year (435 days; questionnaire 5), two years (800 days; questionnaire 6), three years (1165 days; questionnaire 7), and four years (1530 days; questionnaire 8) after surgery. All patients treated for advanced rectal cancer in our department were consecutively invited for participation in this study. A total of 594 patients were asked to participate in this study, and 489 patients completed at least one questionnaire (Figure 1).

Initially, the Non-COVID-19 study group consisted of 328 patients and the COVID-19 group consisted of 61 patients. The discrepancy between the number of patients who completed the first questionnaire and the patient sample size can be explained by the fact that some patients chose to complete the second questionnaire but not the first and vice versa. On average, each patient completed 4.27 questionnaires. Consequently, patients were interviewed consecutively, and data were collected prospectively. Clinical data were collected retrospectively from patient records. Of the 489 surveyed patients, 343 were male and 146 were female. Patients ranged in age from 15 to 93 years at the start of therapy (day 0), with a mean age of 63.5 years. All questionnaires were divided into a Non-COVID-19 and a COVID-19 group, separated by the beginning of March 2020. Therefore, each survey, not patient, was assigned to a group, resulting in some patients changing from the Non-COVID-19 to the COVID-19 group over time. This, in conjunction with varying and also declining response rates, as well as patients’ deaths, led to different sample sizes for each questionnaire before COVID-19 (87 to 328 patients) and during COVID-19 (28 to 63 patients).

As part of the patient sample used in this paper with data extending to June 2021 was examined by another study for the time points one to four [13], we focused on the annual follow-ups, developments throughout all questionnaires—including by sex and age—and the QOL dependent on the clinical T-stage of the cancer. To illustrate the development across all questionnaires, the combined means of all nine functional scores and the combined means of all fourteen symptom scores were compared before and during COVID-19 for all eight questionnaires. Further analyses were performed across all eight time points for subgroups of sex, comparing males to females, and age, comparing patients 65 years and older to patients younger than 65 years. In addition, subgroups were created based on the clinical T-stage of the patients’ rectal cancer and plotted over the course of 70 days, including four questionnaires. A subgroup with less advanced cancer (cT1–cT3) was compared to a subgroup with more advanced cancer (cT4). Since the less advanced cancer subgroup includes mostly cT3-stage cancers, less advanced has to be seen in relative terms.

All patients received their questionnaire either in person during radiotherapy or by mail after completion of treatment. The answers were digitised in Excel 2016 using a written code in Visual Basic for Applications (VBA). The COVID-19 and Non-COVID-19 groups were formed using Excel 2016. Sample sizes were unequal due to the specific time limit of the COVID-19 pandemic. All data analyses, including *p*-values, means, standard deviations, medians, and percentiles were conducted using GraphPad Prism 9.0.2, where *p*-values < 0.05 as well as a difference of 10 or more percentage points (pp) were considered to be statistically significant. Comparisons between Non-COVID-19 and COVID-19 groups were achieved by making use of the Kolmogorov-Smirnov normality test, the Welch’s unequal variances t-test, and graphically plotted via GraphPad Prism 9.0.2.

## 3. Results

A total of 489 advanced rectal cancer patients who were treated with radiochemotherapy were surveyed at eight different time points before and during the COVID-19 time, with the latter one being from March 2020 to March 2022. Results were compared to analyse the effect of the COVID-19 pandemic on the patients’ QOL. For functional scores, higher values indicate better QOL, while for symptom scores, lower percentages are more desirable.

The different characteristics of all patients, including age, sex, grading, and TNM- as well as UICC-stages, are shown in Table 1. Each patient was assigned to the “Non-COVID-19”, “COVID-19”, or for reasons of clarity, the specifically created “Varying Non-COVID-19/COVID-19” column. The latter includes all patients who changed groups over time and completed surveys before and during COVID-19.

All functional scores decreased during the COVID-19 pandemic at the first annual follow-up (Figure 2A) except sexual function (+3.7 pp), with body image falling the most (−10.6 pp), followed by global health status (−8.1 pp, *p* = 0.067), cognitive functioning (−6.9 pp), and role functioning (−5.8 pp). Future perspective remained the worst rated score (47.9% to 45.5%), while sexual function (74.4% to 78.1%) replaced cognitive functioning (78.6% to 71.7%) as the highest rated score. Similarly, symptom scores tended to rise during the COVID-19 pandemic (Figure 2B), with every score except micturition problems (−4.3 pp) increasing. Especially, gastrointestinal tract symptoms (23.1% to 29.5%, *p* = 0.155), financial difficulties (+8.2 pp), dyspnoea (+7.6 pp), and insomnia (+7.1 pp) were rising. Fatigue remained the highest score (35.5% to 41.8%).

There were few changes in functional scores at the two-year follow-up at day 800 (Figure 3A), except global health status (63.7% to 59.0%, *p* = 0.208) and, in contrast to the first follow-up, small increases in social functioning (+3.6 pp) and body image (+3.6 pp), whereas sexual function decreased slightly (−2.7 pp). The trend of decreasing scores continued, with *p*-values ranging from 0.318 to 0.677, as questionnaire seven (Figure 4A) had small but consistently worsening functional scores before and during COVID-19. One exception was sexual function, and the biggest difference was found for social functioning (−6.0 pp). After four years (Figure 5A), the most pronounced declines were in future perspective (−9.0 pp), role functioning (−7.2 pp), and body image (−7.0 pp). Throughout all follow-ups, the difference in global health status became gradually smaller (8.1 pp → 4.7 pp → 4.2 pp → 1.3 pp).

Symptom scores for the follow-ups mostly rose in the COVID-19 cohort. Most prominent examples for the second follow-up (Figure 3B) were diarrhoea (+10.0 pp), financial difficulties (+8.0 pp), as well as constipation (10.5% to 17.4%, *p* = 0.136), gastrointestinal tract symptoms (21.0% to 26.2%, *p* = 0.124), chemotherapy side effects (15.5% to 21.0%, *p* = 0.131), and defecation problems (30.7% to 40.0%, *p* = 0.075). At the third annual survey (Figure 4B), patients reported the highest changing values for defecation problems (+13.5 pp, *p* = 0.086), insomnia (+10.2 pp), weight loss (+9.8 pp, *p* = 0.036), and gastrointestinal tract symptoms (18.9% to 25.9%, *p* = 0.059). Other scores like pain and dyspnoea remained unchanged or even slightly deteriorated. For follow-up number four (Figure 5B), diarrhoea improved during COVID-19 (−13.5 pp), whereas defecation problems appeared to get significantly worse (28.3% to 39.6%, *p* = 0.089).

Changes in all surveys over time were recorded and summarised in the following figure (Figure 6). In total, every single functional score during COVID-19 performed worse than its Non-COVID-19 counterpart, whereas every symptom score rose during COVID-19, meaning all average scores worsened after March 2020. The average functional score was lowest at 58.2% and 58.1% on day 35, both prior to and after the beginning of the pandemic, with the difference of 0.1 pp being the smallest out of all averages. The highest point was reached at the fourth annual follow-up, where the scores reached their peak at 68.7% before and at 64.8% during COVID-19, which was only surpassed by 65.1% on day 800 during COVID-19. On day 435, there was a difference of 5.0 pp between the two measurements, which was the largest deviation. The results were the opposite for the symptom scores, but with similar implications for the scores. At five weeks, the highest scores for both groups were 36.6% before and 37.4% during COVID-19. The scores of day 35 were also the closest, with a difference of only 0.8 pp. Prior to the pandemic, the second follow-up on day 800 scored the lowest at 20.4%, while throughout COVID-19 it was follow-up four at 23.5%. The largest difference was 4.7 pp at follow-up two.

By breaking down the results of the questionnaires over time, all nine functional scores and all fourteen symptom scores can be analysed individually (Figure 7). Regarding functional scores, role functioning (B) showed a significant difference at day 0 (10.7 pp), followed by the Non-COVID-19 graph approaching the COVID-19 graph (day 35 and 800) and again ending with a greater gap in between on day 1530 (7.2 pp). Body image (G) showed a huge drop from questionnaire four (62.7%) to questionnaire five (52.2%) during the pandemic, bouncing right back (65.0%, questionnaire six) and even surpassing the Non-COVID score (61.4%), marking an outlier in the data. Sexual function (H) was the only score to perform slightly better after March 2020 (except for day 800), with score differences changing between 2.4 and 7.5 pp. Questionnaires one to six were very even for future perspective (I), with the two graphs being separated by 2.4 pp at most; however, the last two surveys showed a continued increase (49.2% → 53.2% → 55.4%) before the pandemic compared to stagnating values during COVID-19 (48.6% → 49.6% → 46.4%) resulting in a difference of 9.0 pp at day 1530.

Symptom scores tended to rise for the first three surveys, reaching their peak at day 35 and then descending again. Pain (L) was found to be similar or higher during the pandemic, except for said peak on questionnaire three (3.9 pp lower). The gaps in between scores for dyspnoea (M) were large, especially at the end of radiotherapy (9.3 pp) and before surgery (12.3 pp). Values for constipation (P) seemed to be mirrored; day 14 offered the highest (Non-COVID: 19.7%) and second lowest (COVID: 12.9%) score, while day 435 showed the third lowest (Non-COVID: 12.0%) and highest (COVID: 17.7%) scores. Financial difficulties (R) in the Non-COVID-19 group were reported to only differ to some degree (18.8% to 26.3%, 7.5 pp) over time, whereas in the COVID-19 group compared to the Non-COVID group the first four scores were lower by a margin of 4.4 to 13.9 pp, questionnaire five and six rapidly rising to 31.3% (8.0 and 8.2 pp higher) and then dropping back to 21.7% and 23.8% for surveys seven and eight, offering a wide variety of values. Defecation problems (V) differed by 1.1 to 5.8 pp for the first five surveys, while the last three surveys showed larger differences of 9.3 to 13.5 pp.

In addition to analysing the impact of the pandemic on all patients, we attempted to divide the patient sample size by sex (Figure 8) and age (Figure 9). For females (Figure 8A), the first four functional scores differed by a margin of only −0.2 (= COVID-19 score better by 0.2 pp) to 1.0 pp, while the annual follow-ups differed by at least 4.2 pp. Day 435 with 62.1% and 45.3% (16.8 pp) and day 1530 with 66.0% and 54.1% (11.9 pp) showed the largest gaps. The lowest functional score before COVID-19 was 52.0% at week 5 and 45.3% at day 435 during COVID-19, with the highest at the fourth annual follow-up before the pandemic (66.0%) and at the second annual follow-up during the pandemic (60.1%). On the other hand, slightly larger gaps were found for symptom scores during the first ten weeks (0.2 to 4.5 pp) and slightly smaller gaps at day 435 with 16.4 pp and at day 1535 with 9.4 pp. The highest and lowest symptom scores were at day 35 and 800, respectively, with 42.2% and 23.8% before COVID-19 and 42.9% and 29.3% during COVID-19. In general, functional and symptom score trends were similar.

The functional score results for males (Figure 8B) did not show such large differences; they were between −1.2 and 4.0 pp. During COVID-19, days 35, 435, and 800 all outperformed their Non-COVID-19 counterparts. The highest scores were surveyed at annual follow-up number four with 70.0% prior to COVID-19 and at annual follow-up number two with 67.7% during COVID-19, whereas the lowest scores were surveyed at day 35 (60.5% before and 61.5% during the pandemic). Symptom scores during COVID-19 were slightly lower at day 35 (0.2 pp) and at day 435 (2.4 pp). The largest difference of 3.8 pp was found at day 800. Pre-pandemic scores ranged from 19.1% on day 1530 to 34.5% on day 35, while scores during the pandemic ranged from 19.4% on day 435 to 34.3% on day 35. Women reported lower QOL than men in all surveys, regardless of the pandemic.

For patients aged 65 years or older (Figure 9A), the functional score at week 5 during COVID-19 was better by a margin of 1.5 pp; all other scores were outperformed by the Non-COVID-19 time, with the largest margin being 5.3 pp on day 1165. Here, 58.8% and 60.3% on day 35 both appeared to be the worst scores across all eight questionnaires, resulting in day 35 before COVID-19 being the lowest of all functional scores for patients aged 65 years or older. Peak scores were reached on day 1165 with 71.4% prior to the pandemic and on day 1530 with 66.3% during the pandemic. Meanwhile, during COVID-19, symptom scores performed better on days 14, 35, and 435. All ranges were between −2.9 and 5.6 pp. The lowest symptom score before COVID-19 was 18.6% at the third annual follow-up and 21.4% during COVID-19 at the first annual follow-up. The highest symptom scores were surveyed at the third questionnaire (37.0% before and 34.1% during the pandemic).

In contrast, patients aged younger than 65 years (Figure 9B) showed greater differences in QOL in the first four surveys, ranging from 2.6 to 5.1 pp for functional scores and from 4.7 to 6.9 pp for symptom scores. Between functional scores, the largest difference was 5.1 pp on day 35 and between symptom scores, it was 8.2 pp on day 435. Here, 66.0% four years after treatment before the pandemic and 64.6% two years after treatment during the pandemic outperformed all other functional scores, while day 35 was the lowest at 57.8% before and at 55.2% during COVID-19. Correspondingly, symptom scores peaked at day 35 (36.2% prior to and 41.7% during COVID-19). The lowest symptom scores were 21.8% at the first follow-up and 24.8% at the fourth follow-up, respectively.

For the first four questionnaires, a further subdivision was made based on the clinical T-stage of the patients’ rectal cancer. Thus, patients with less advanced cancer (cT1–cT3; subgroup 1) were compared to patients with more advanced cancer (cT4; subgroup 2) (Figure 10 and Figure 11). The first score, physical functioning (A), decreased similarly (between 5.7 and 8.8 pp) for the less advanced rectal cancer patients, whereas the cT4-subgroup decreased slightly (2.3 and 2.6 pp), followed by higher increases (4.0 and 6.4 pp) for the third and fourth surveys. This development was similar for role functioning (B). Emotional functioning (C) was just slightly affected by the pandemic, irrespective of cancer stage (up to 3.5 pp, one outlier on day 35 at 5.6 pp). Cognitive (D) and social (E) functioning showed significant differences at the end of radiotherapy (10.2 and 10.0 pp) for the higher cT-stage subgroup, in contrast to small differences for the lower cT-stage group (differences of 2.5 pp or less). While body image dropped during COVID-19 for subgroup 2, there was a small rise during (5.3 pp) and after (5.4 pp) radiotherapy for subgroup 1. Sexual function (H) for cT4-stage cancer patients rose significantly before (15.8 pp to 90.3%) and during (11.6 pp to 90.5%) radiotherapy, surpassing the increases of subgroup 1 by a margin. The highest score regarding future perspective (I) was obtained in the second survey for subgroup 2 regardless of COVID-19, exceeding all other COVID-19 patient scores by at least 6 pp.

There was a significant difference of 10.6 pp for fatigue (J) for subgroup 1 on day 0, and a huge gap of 9.1 and 8.7 pp on day 14 across all cancer stages. Nausea and vomiting (K) had opposing tendencies: while all COVID scores got higher for the cT1-3 patient cohort (day 0: 6.3% to 16.3%), the cT4 pandemic scores showed lower results except for day 0 (+0.9 pp). Flattened score developments for pain (L) were surveyed during COVID-19, where the pre-COVID-19 peak of 47.3% and 47.6% was missed by 3.4 and 5.4 pp on week 5. Dyspnoea (M) displayed differences of 8.6 pp and above for all scores for subgroup 1, with the jump from 24.4% to 40.7% on day 70 being the largest. The T4 subgroup, on the other hand, revealed significant gaps in both directions (26.8% to 15.6% on day 0, 26.2% to 37.8% on day 35). Whereas results regarding appetite loss (O) for subgroup 1 differed by 4.4 pp or less, subgroup 2 offered significantly higher results on day 0 (10.8 pp) and day 35 (11.4 pp). Diarrhoea (Q) showed increased results for all scores, with day 0 of subgroup 2 being significant (11.5 pp). Conversely, scores for financial difficulties (R) were all lower, with four significant differences (subgroup 1: at day 35 −13.7 pp; subgroup 2: at day 0 −10.7 pp, at day 35 −15.6 pp, and at day 70 −13.9 pp). Micturition problems (S) differed by 9.4 and 9.6 pp on days 14 and 35 for subgroup 2. The pandemic had a greater effect on defecation problems (V) in subgroup 2, with differences ranging from 4.2 to 6.6 pp, whereas in subgroup 1 the differences ranged from −0.3 to 3.7 pp, except on day 70, when it was higher (6.3 pp). Weight loss (W) for more advanced cancer patients was affected by COVID-19 remarkably, with scores rising between 10.7 and 13.7 pp; however, the less advanced cohort showed fewer differences ranging from −3.7 to +7.9 pp.

## 4. Discussion

During the COVID-19 pandemic, rectal cancer patients’ QOL declined, and functional scores such as role functioning and body image were impaired. This, in addition to the rising perception of symptoms such as insomnia, defecation problems, and weight loss, led to the results. At one to four years of follow-up, few clear differences were found, but most scores worsened, indicating a negative impact of COVID-19 on QOL. This trend was also evident when all functional and symptom scores were combined for comparison. The differences between COVID-19 and Non-COVID-19 were smallest at the end of radiochemotherapy, when QOL was most affected. The individual scores were very diverse, with some of them diverging during the radiochemotherapy (dyspnoea), even years after the therapy (future perspective), or on the contrary improving (sexual function). Sex had an impact, with females reporting worse QOL than males in all surveys before and during COVID-19. Age also had an influence, as patients younger than 65 were more affected by the pandemic during radiochemotherapy than their older counterparts. Scores for TNM-stages cT1-3 and cT4 did not change uniformly; however, weight loss was significantly higher during COVID-19 for the latter group.

A clear limitation is that the results were only compared between rectal cancer patients and not with the general population. Vulnerable adult populations have experienced mental health impairment due to physical distancing, stay-at-home orders, and other lockdown restrictions [11,14]. Therefore, the additional significance of the rectal cancer diagnosis remains unclear. Countermeasures for all adults should be applied to rectal cancer patients as well, thus reducing the need for specific measures.

Since voluntary questionnaires can be declined by patients at any time, there is a selection bias in the study group. As our surveys were all delivered by post, COVID-19 may have contributed, leading to former participants not completing surveys. Excessively anxious patients who do not participate would produce biased and embellished results. Alternatives such as digital questionnaires might be used in the future, eliminating some COVID-19-related concerns; however, adding limits for digital immigrants, as the median age of rectal cancer diagnosis in developed countries is around 70 years [15]. Viral infections of COVID-19 may also cause symptoms similar to some of those listed in the questionnaire and thus affect the results. During the six weeks of radiochemotherapy, there were no cases of COVID-19. However, we do not have these data for further follow-up and cannot exclude bias due to possible COVID-19 disease. Another limiting concern is the length of our study. The COVID-19 group included patients over a two-year time course and varying restrictions. A total of fifteen Bavarian Infection Protection Measures Ordinances, not including amendments, had been in force in that time span. COVID-19 vaccines were available since December 2020. This presumably led to QOL changes, which were not included in the study design. We decided not to use a shorter time period because a sample size of approximately 30 patients for each annual follow-up seemed appropriate. Nonetheless, comparisons for the first four questionnaires were published by us for a briefer time period (March 2020–June 2021) [13].

Strengths of our study include its prospective nature, in which changes in functioning and symptom perception were frequently assessed during the different phases of therapy. Therefore, changes could be accurately measured. In addition, the large number of patients over more than a decade resulted in a great patient sample and thus a large amount of study data. Questionnaires several years after therapy represent a long follow-up period and support this on an individual basis.

The decline in almost all functional scores one to four years after radiotherapy revealed that long-term rectal cancer patients’ QOL suffered. Physical activities decreased during COVID-19 [16], which can have an impact on overall physical functioning and QOL in older adults [17]. This may explain the worsening at all follow-ups. As everyday life changed during the pandemic, remote work emerged and outdoor activities were limited, which provides an explanation for diminished role functioning scores. With each year of follow-up, the gap between the two global health status scores narrowed, suggesting that the impact of the pandemic on general QOL and health decreased over time. However, thoughts regarding the future perspective diverged more over time. Possibly due to still facing the COVID-19 global health crisis, patients with a medical history of rectal cancer did not report increased scores, unlike their pre-pandemic counterparts. The risk of getting infected with COVID-19 and being more vulnerable to it because of said medical history could be the reason for this. The decline in body image might be a result of patients not being able to feel what their bodies are capable of, having to stay at home, and the fear of gaining weight. Though this does not explain why the pandemic group had higher scores at follow-up two after plummeting in year one, both seeming to be outliers. Sexual function was an exception to the decreasing scores, as it increased in three out of four follow-ups. As a consequence of stay-at-home orders during the pandemic, outside activities were limited and the time spent at home rose. This might have led to more situations craving and actually engaging in sexual activities, thus leading to higher sexual function.

Of all the symptom scores, dyspnoea offered the most apparent changes. As it is also a symptom of COVID-19, an infection during answering the questionnaire might have led to higher scores besides also sensitising patients to the symptom itself without them getting sick. Feeling weaker during radiochemotherapy may contribute to this; music therapy could be a countermeasure [18]. Worries about the pandemic led to higher rates of insomnia in the general population [19], which is also clearly evident in our study. Diarrhoea and gastrointestinal tract symptoms are common symptoms after surgery for colon cancer [20], but can also be caused by COVID-19. Thus, elevated scores are likely due to infection or higher symptom awareness. The differences in diarrhoea at four years in favour of the COVID-19 group, however, contradict this. Financial difficulties during the first ten weeks of treatment may have been lower, as patients were more concerned about their health, just having received a cancer diagnosis and hence being more susceptible to SARS-CoV-2. Constant contact with mask-wearing health care workers due to daily radiotherapy may have contributed to those thoughts, resulting in financial difficulties not being a primary concern, although financial distress is common among cancer patients undergoing radiotherapy [21]. A few years after treatment, getting used to the diagnosis as well as not visiting hospitals regularly could lead to more present financial thoughts before adapting to that. Given that patients stayed at home more during the pandemic, defecation problems could have been more apparent for cancer patients in general, as well as a consequence of a SARS-CoV-2 infection. Staying at home and less physical activity may lead to an increase in weight, yet patients reported weight loss. Long-term patients might have skipped follow-up examinations and hence not detected tumour recurrences. Other reasons might be increased anxiety and COVID-19 infections [22].

Both before and during COVID-19, QOL scores were reported to be lowest and closest after radiochemotherapy (day 35). This may be because patients’ already compromised QOL is more stable to external factors such as the pandemic, as their main focus is on their disease and the immediate side effects of radiochemotherapy [3,4], superseding possible threats from COVID-19.

Regardless of the pandemic, females had lower QOL than males in all surveys, which is supported by our findings from March 2020 to June 2021 for the first ten weeks of treatment [13]. During treatment, females receive slightly higher deposited energies per weight of ionising radiation, which could explain the differences in QOL [23]. In addition, increased chemotherapy-induced toxicity in women could worsen QOL [24]. The annual follow-ups of females stood out, as their QOL decreased by a great margin during the pandemic. Sex-related factors are important, as females who are pregnant, have had a miscarriage, are postpartum, or experience physical abuse are at greater risk of mental health impairments [25]. However, this does not explain the huge differences in QOL years after treatment. Though, it should be noted that the follow-ups were completed by only five to sixteen females, limiting the findings.

Patients who were younger than 65 years of age were more affected by the COVID-19 pandemic during radiochemotherapy than patients aged 65 years or older. Some studies have pointed out that the impact of radiochemotherapy on QOL decreases with increasing age [26,27]. The COVID-19 pandemic or other external factors may be an additional burden for younger patients during treatment, while older patients are less affected by it. Another study found that increasing age is a protective factor for mental health during COVID-19, as it is associated with less stress and fewer psychological problems [28]. This may be due to life reflection and adaptive use of personal memory and generativity [29], and highlights the resilience of older patients.

Compared to the time before COVID-19, cT4-staged rectal cancer patients lost more weight during the pandemic than their cT1-3 counterparts, which is directly linked to higher appetite loss. COVID-19 infections and worries could have contributed to this. Delays in seeking medical care may have led to more frequent proliferation of cancer cells. Because of their exponential growth, this would have had a more consuming effect on patients with higher stages of cancer, explaining the findings.

Research using the QLQ-C30 questionnaire and comparing QOL prior to and during the pandemic revealed different results: A study observing Danish cancer patients [30] showed comparable results, though lower QOL and the fear of getting infected with COVID-19 were correlated. Another survey from Denmark investigating hematologic cancer patients indicated lower QOL during the pandemic [31]. Among Italian head and neck cancer survivors, a significant deterioration in physical (*p* = 0.028), role (*p* = 0.030), and emotional functioning (*p* = 0.041) was found [32]. Significantly worse insomnia (*p* < 0.0001), fatigue (*p* = 0.003), loss of appetite (*p* = 0.006), financial difficulties (*p* < 0.0001), and cognitive (*p* < 0.0001) as well as social functioning (*p* < 0.0001) scores were found for stage III/IV cancer patients in Poland [33]. However, one should be cautious when transferring results to this study. Besides either including other or different types of cancer and covering shorter time periods, countries in Europe showed varying COVID-19 developments [34]. A study conducted by us regarding an overlapping rectal cancer patient sample for the first four questionnaires from March 2020 to June 2021 showed more significances between the COVID-19 and Non-COVID-19 groups. Particularly, day 0 stood out with five functional (physical, role, cognitive, social functioning, global health status) and four symptom scores (fatigue, pain, nausea and vomiting, weight loss) changing significantly, though the differences decreased gradually over the next time points [13].

This study focused on the impact of the pandemic on rectal cancer patients. During a pandemic, there is a need to be more responsive to patients and their concerns and fears, as the crisis has an additional negative impact on QOL on top of the catastrophe of the disease itself. However, patients’ QOL may be affected not only by the COVID-19 pandemic, but also by other similar events such as economic, political, or environmental crises. Patients should be offered more psychological support during such periods.

What we cannot assess is whether the pandemic itself, through the threat of COVID-19, or the restrictions imposed by policymakers, or both, caused the decline in QOL. We are even less able to assess whether fewer restrictions would have helped to improve QOL.

## 5. Conclusions

Trends in impaired quality of life were observed in long-term rectal cancer patients during COVID-19, even years after they became diagnosed with cancer. External influences such as the COVID-19 pandemic can have a significant impact on quality of life in addition to cancer and the effects of its treatment. One could speculate that such serious events as the war in Ukraine will also affect the quality of life. This should be considered when providing psychological care.

## Figures and Tables

**Figure 1 healthcare-11-01981-f001:**
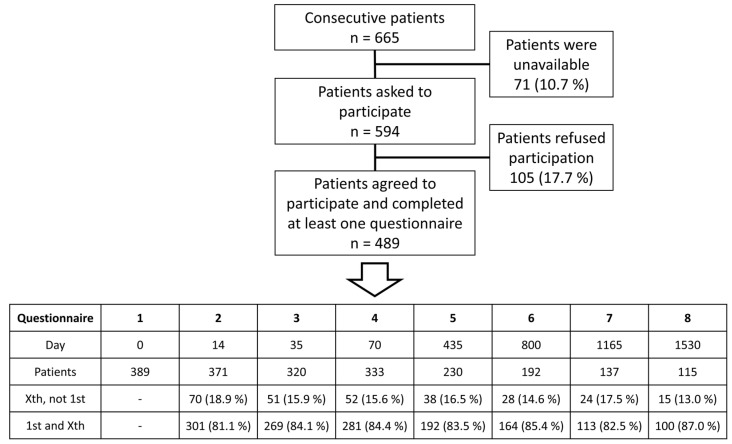
Flowchart of patients leading to the 489 study participants. The table shows the number of patients per questionnaire and is further divided into patients who completed a subsequent questionnaire without completing the first initial questionnaire and patients who completed both the first and a subsequent questionnaire.

**Figure 2 healthcare-11-01981-f002:**
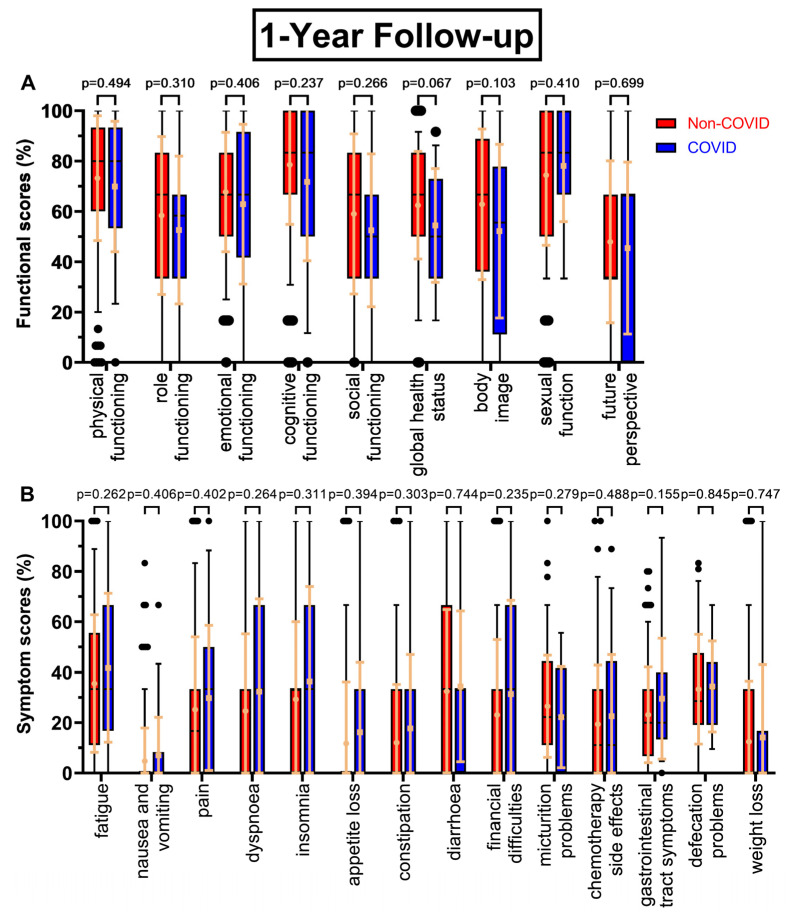
Boxplots of functional and symptom scores prior to (Non-COVID) compared to during (COVID) the COVID-19 pandemic, with whiskers defined as the 5th and 95th percentile and points marking outliers. Means and standard deviations are shown over the boxplots in light orange. If no boxplots are displayed, more than 75% of the values are at 0%. Scores were collected one year after surgery. Functional scores can be seen in (**A**), symptom scores in (**B**).

**Figure 3 healthcare-11-01981-f003:**
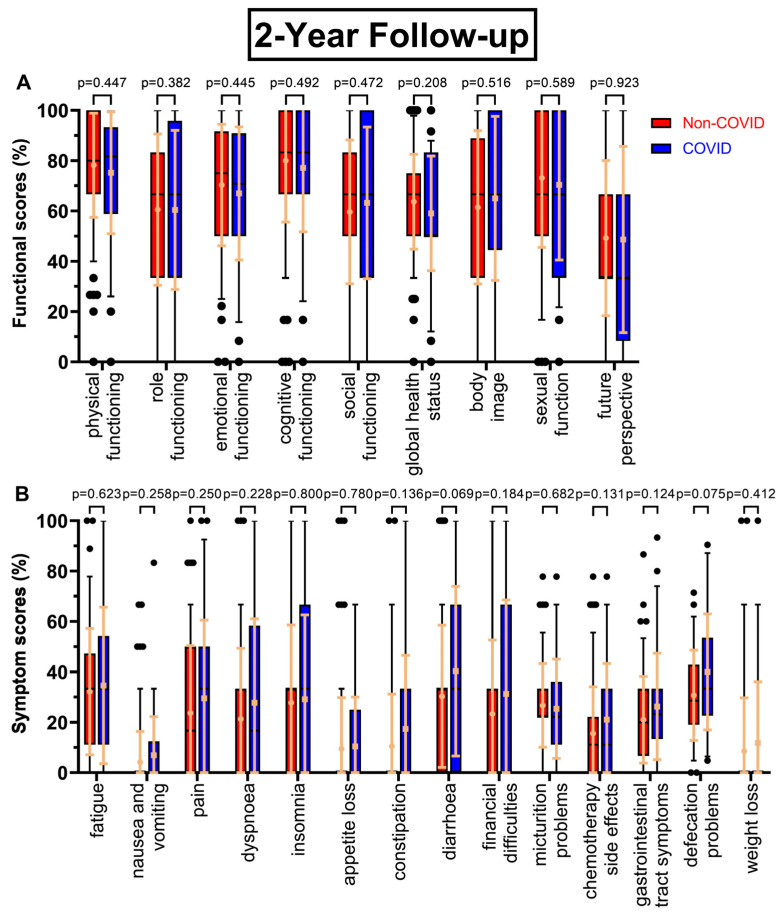
Boxplots of functional and symptom scores prior to (Non-COVID) compared to during (COVID) the COVID-19 pandemic, with whiskers defined as the 5th and 95th percentile and points marking outliers. Means and standard deviations are shown over the boxplots in light orange. If no boxplots are displayed, more than 75% of the values are at 0%. Scores were collected two years after surgery. Functional scores can be seen in (**A**), symptom scores in (**B**).

**Figure 4 healthcare-11-01981-f004:**
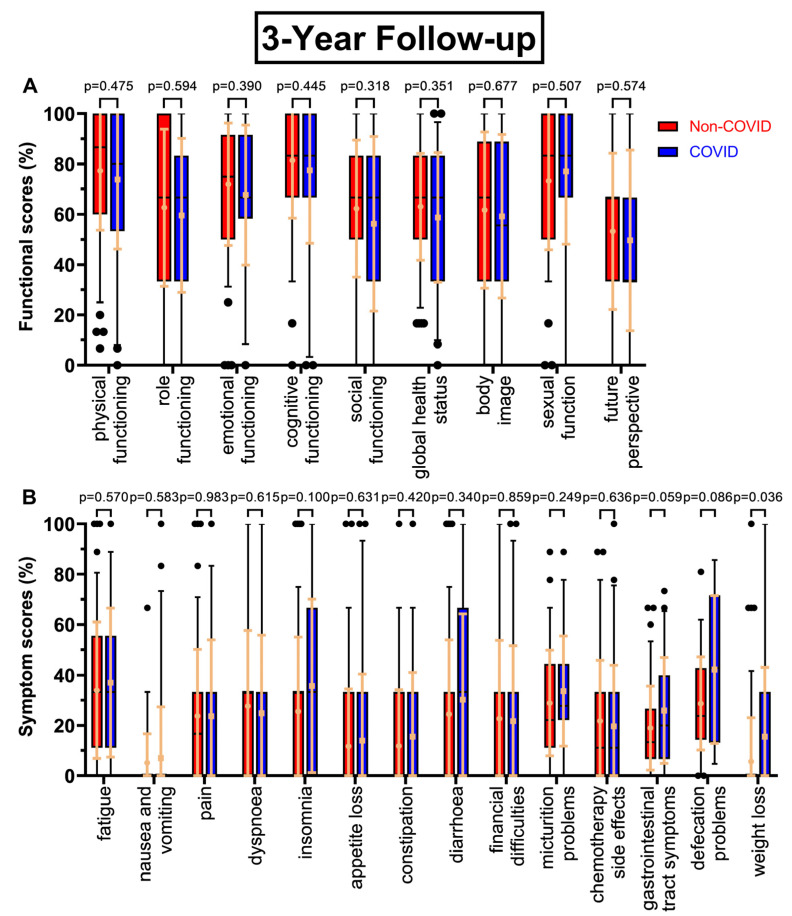
Boxplots of functional and symptom scores prior to (Non-COVID) compared to during (COVID) the COVID-19 pandemic, with whiskers defined as the 5th and 95th percentile and points marking outliers. Means and standard deviations are shown over the boxplots in light orange. If no boxplots are displayed, more than 75% of the values are at 0%. Scores were collected three years after surgery. Functional scores can be seen in (**A**), symptom scores in (**B**).

**Figure 5 healthcare-11-01981-f005:**
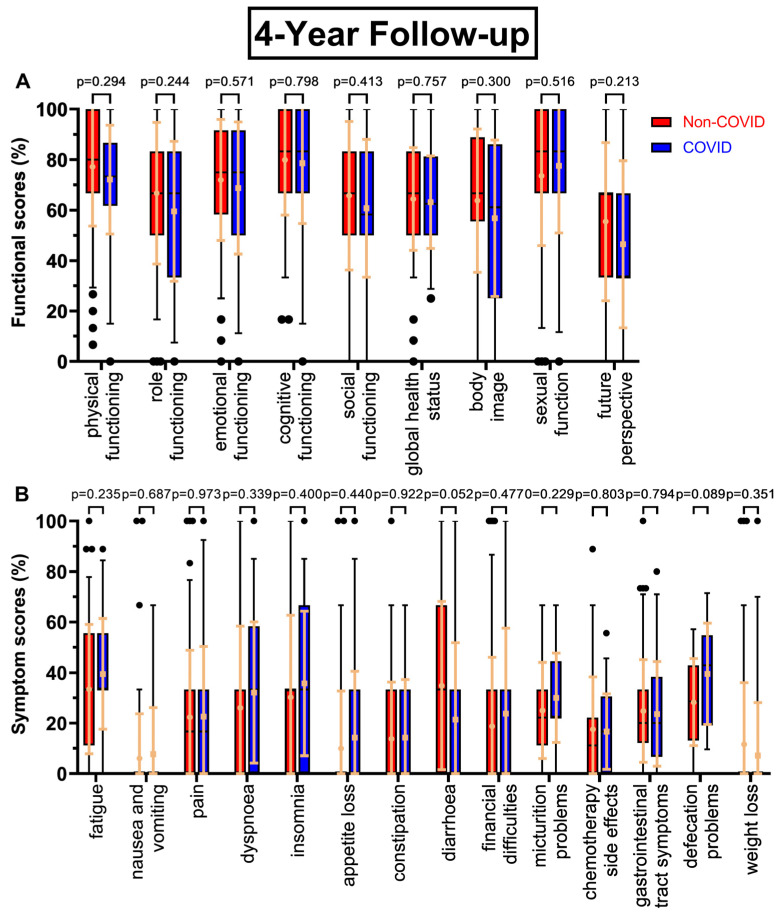
Boxplots of functional and symptom scores prior to (Non-COVID) compared to during (COVID) the COVID-19 pandemic, with whiskers defined as the 5th and 95th percentile and points marking outliers. Means and standard deviations are shown over the boxplots in light orange. If no boxplots are displayed, more than 75% of the values are at 0%. Scores were collected four years after surgery. Functional scores can be seen in (**A**), symptom scores in (**B**).

**Figure 6 healthcare-11-01981-f006:**
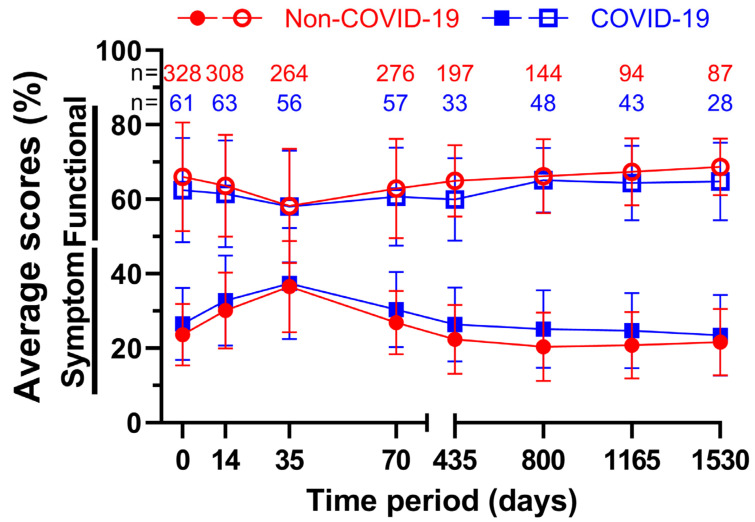
Means and error bars of nine functional and fourteen symptom scores combined, with sample sizes. Functional scores have open symbols and symptom scores have filled symbols. Prior to (Non-COVID) compared to during (COVID) the COVID-19 pandemic over the course of eight time points: before (day 0), during (day 14) and at the end (day 35) of radiotherapy, right before surgery (day 70), and at yearly intervals after surgery (days 435, 800, 1165, and 1530).

**Figure 7 healthcare-11-01981-f007:**
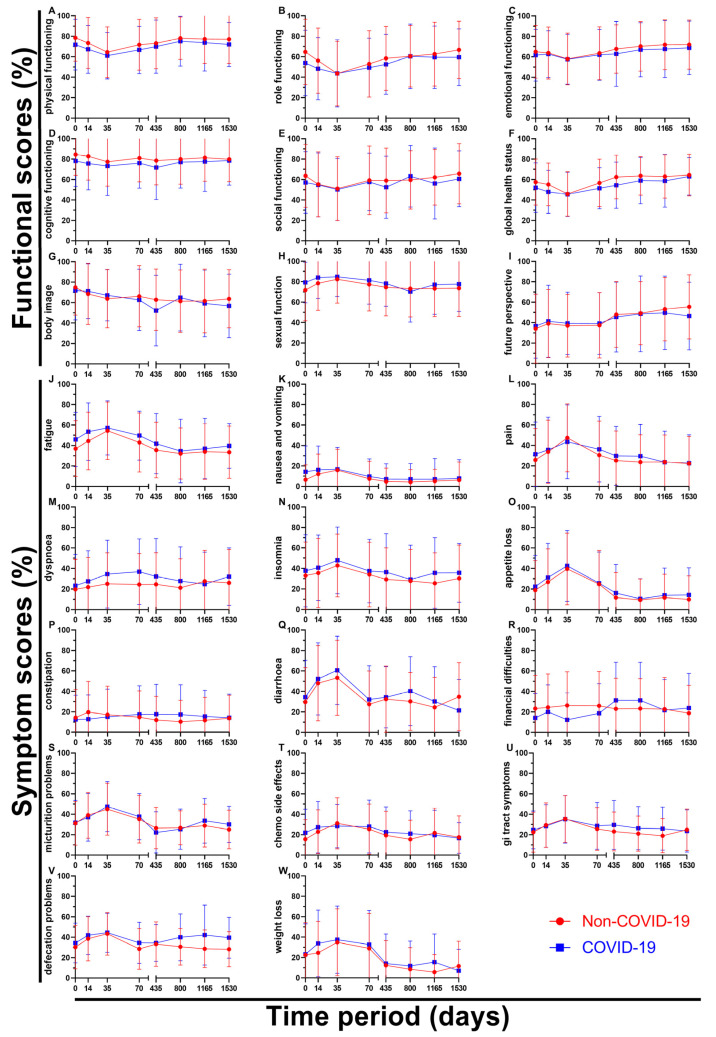
Means and error bars of nine functional (**A**–**I**) and fourteen symptom scores (**J**–**W**). Prior to (Non-COVID) compared to during (COVID) the COVID-19 pandemic over the course of eight time points: before (day 0), during (day 14) and at the end (day 35) of radiotherapy, right before surgery (day 70), and at yearly intervals after surgery (days 435, 800, 1165, and 1530). Each letter represents the score written to the left of its subfigure.

**Figure 8 healthcare-11-01981-f008:**
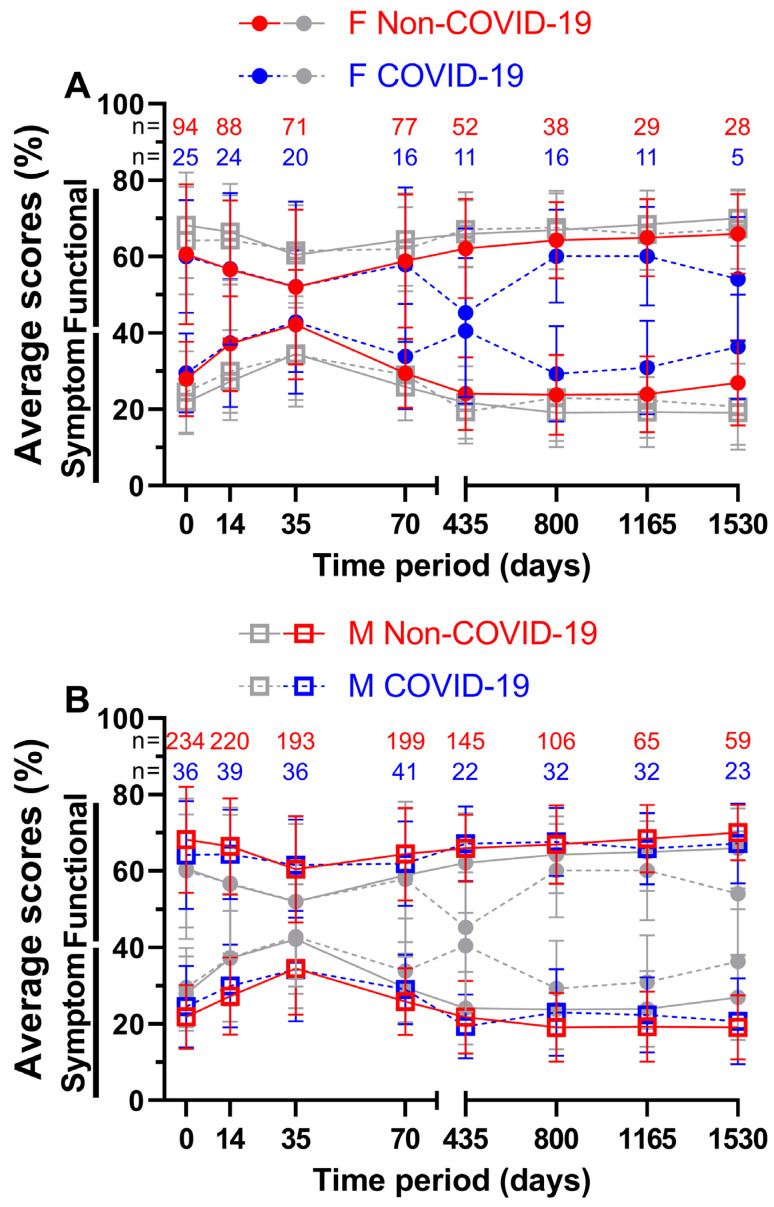
Means and error bars of nine functional and fourteen symptom scores combined, by sex and with sample sizes. Prior to (Non-COVID) compared to during (COVID) the COVID-19 pandemic over the course of eight time points: before (day 0), during (day 14) and at the end (day 35) of radiotherapy, right before surgery (day 70), and at yearly intervals after surgery (days 435, 800, 1165, and 1530). Females are highlighted (grey symbols in the background indicate males) in (**A**), males (grey symbols in the background indicate females) in (**B**). Solid lines are Non-COVID and dashed lines are COVID scores.

**Figure 9 healthcare-11-01981-f009:**
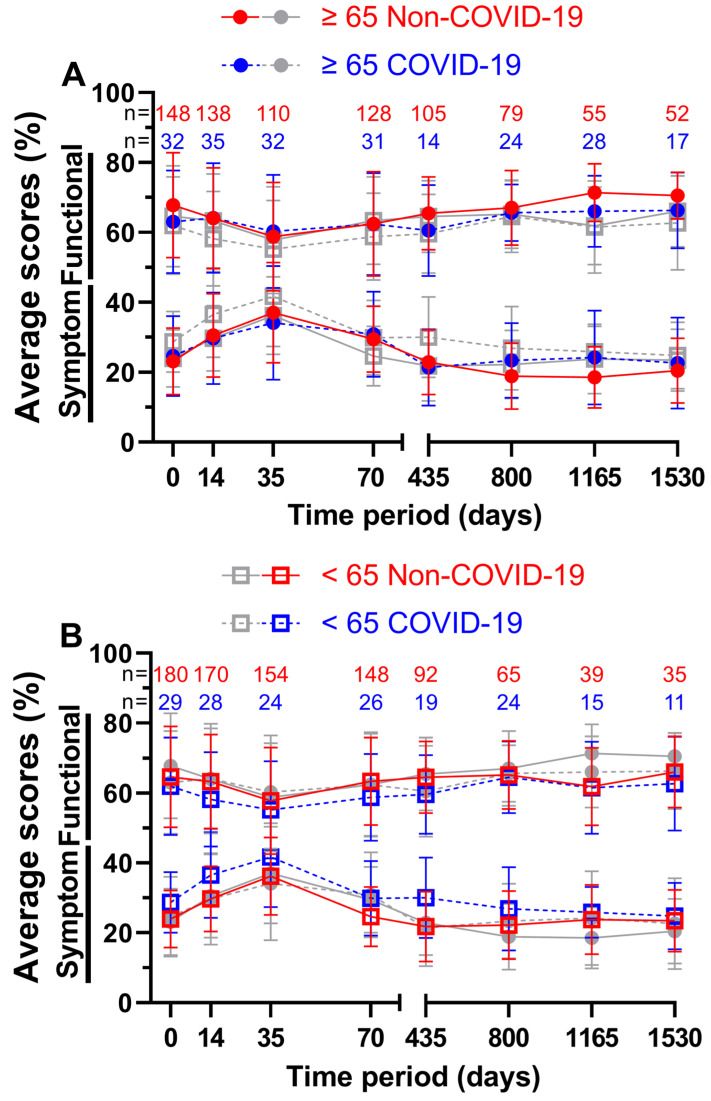
Means and error bars of nine functional and fourteen symptom scores combined, by age and with sample sizes. Prior to (Non-COVID) compared to during (COVID) the COVID-19 pandemic over the course of eight time points: before (day 0), during (day 14) and at the end (day 35) of radiotherapy, right before surgery (day 70), and at yearly intervals after surgery (days 435, 800, 1165, and 1530). Patients aged ≥ 65 years are highlighted (grey symbols in the background indicate patients aged < 65 years) in (**A**), patients aged < 65 years (grey symbols in the background indicate patients aged ≥ 65 years) in (**B**). Solid lines are Non-COVID and dashed lines are COVID scores.

**Figure 10 healthcare-11-01981-f010:**
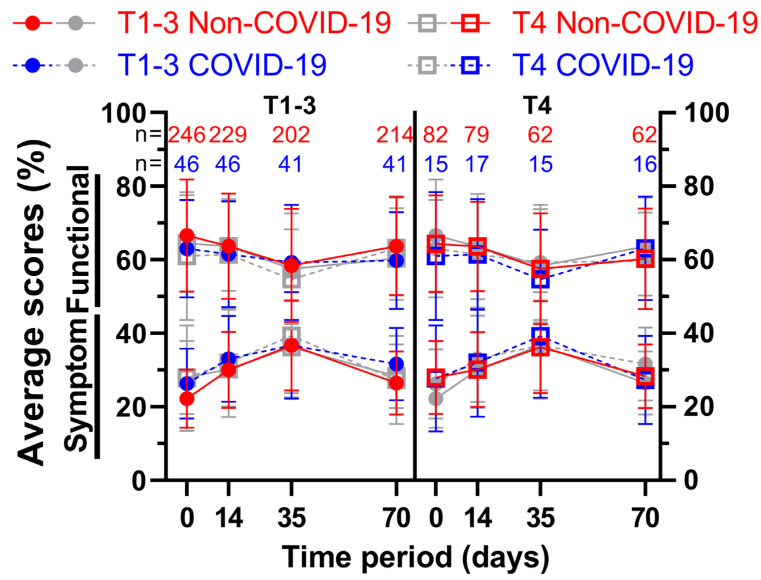
Means and error bars of nine functional and fourteen symptom scores combined, by cT-stage and with sample sizes. Prior to (Non-COVID) compared to during (COVID) the COVID-19 pandemic. Each side shows four graphs with the same four data sets, with the left side putting emphasis on (=colouring) less advanced cancers (TNM cT1-3), the right side on more advanced cancers (TNM cT4). The data are plotted over the course of four time points: before (day 0), during (day 14) and at the end (day 35) of radiotherapy, and right before surgery (day 70). Solid lines are Non-COVID and dashed lines are COVID scores.

**Figure 11 healthcare-11-01981-f011:**
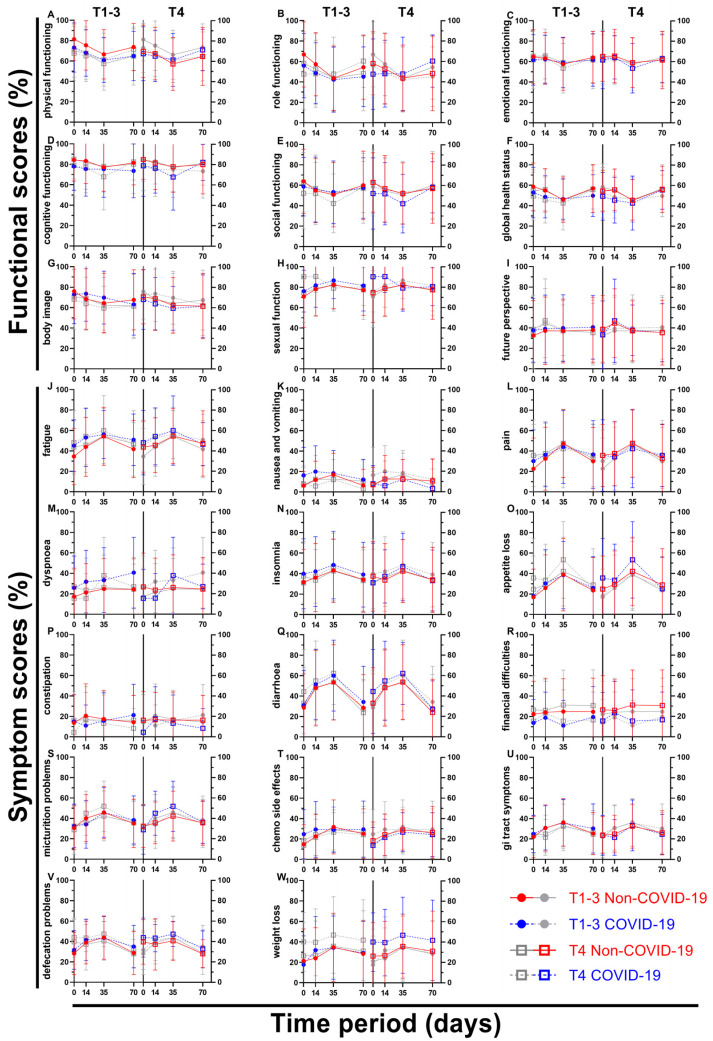
Means and error bars of nine functional (**A**–**I**) and fourteen symptom scores (**J**–**W**). Prior to (Non-COVID) compared to during (COVID) the COVID-19 pandemic. Each letter displays one subfigure with two graphs and represents the score written to the left of its subfigure. All graphs show the same four data sets, with the left side of a subfigure putting emphasis on (=colouring) less advanced cancers (TNM cT1-3), the right side on more advanced cancers (TNM cT4). The data are plotted over the course of four time points: before (day 0), during (day 14) and at the end (day 35) of radiotherapy, and right before surgery (day 70).

**Table 1 healthcare-11-01981-t001:** Descriptive statistics of the surveyed group of patients.

	Non-COVID-19 (%)					Varying Non-COVID-19/COVID-19 (%)					COVID-19 (%)				
Patients	244 (50)					169 (35)					76 (16)				
Age	mean: 63.5 years		range: 23–86 years			mean: 63.1 years		range: 15–90 years			mean: 64.3 years		range: 41–93 years		
Sex	male: 174 (71)		female: 70 (29)			male: 122 (72)		female: 47 (28)			male: 47 (62)		female: 29 (38)		
Stage	0 (%)	1 (%)	2 (%)	3 (%)	4 (%)	0 (%)	1 (%)	2 (%)	3 (%)	4 (%)	0 (%)	1 (%)	2 (%)	3 (%)	4 (%)
cT	-	1 (0)	25 (10)	153 (63)	65 (27)	-	4 (2)	12 (7)	113 (67)	40 (24)	-	2 (3)	10 (13)	45 (59)	19 (25)
pT	26 (11)	18 (7)	71 (29)	106 (44)	24 (10)	30 (18)	15 (9)	39 (23)	72 (43)	13 (8)	8 (11)	7 (9)	20 (26)	39 (51)	3 (4)
cN	62 (25)	128 (53)	54 (22)	-	-	40 (24)	71 (42)	58 (35)	-	-	17 (23)	31 (41)	27 (36)	-	-
pN	152 (62)	58 (24)	34 (14)	-	-	130 (77)	31 (18)	8 (5)	-	-	57 (75)	15 (19)	4 (5)	-	-
cM	190 (78)	54 (22)	-	-	-	149 (88)	20 (12)	-	-	-	58 (77)	18 (23)	-	-	-
cUICC	-	10 (4)	43 (18)	136 (56)	55 (23)	-	8 (5)	28 (17)	111 (66)	21 (13)	-	4 (6)	10 (13)	44 (58)	18 (24)
pUICC	-	60 (25)	74 (30)	72 (30)	38 (16)	-	58 (34)	58 (34)	35 (21)	19 (11)	-	21 (28)	24 (32)	14 (19)	17 (22)
Grading	-	9 (4)	194 (80)	41 (17)	-	-	9 (5)	137 (81)	23 (14)	-	-	2 (3)	62 (82)	12 (16)	-

Non-COVID-19 = patients interviewed exclusively before the COVID-19 period. Varying Non-COVID-19/COVID-19 = patients interviewed before and during the COVID-19 period. COVID-19 = patients interviewed exclusively during the COVID-19 period.

## Data Availability

The data that support the findings of this study are available from the corresponding author upon reasonable request.

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
