# Peer review of "Impact of COVID-19 on Quality of Life in Long-Term Advanced Rectal Cancer Survivors"

_healthcare, 2023, doi:10.3390/healthcare11141981_

Round 1

Reviewer 1 Report

Thanks for giving me the chance to review this manuscript. I really enjoyed its reading .

The introduction is clear and to the point.

For the method,Did the authors conduct any validation for the survey?

Was the study registered in ClinicalTrials.gov or approved by ethics'committee? The authors should write the registration number or the number of ethical approval indicating the ethics'committee

Did the patients sign the informed consent to be included in the study ?

The results are very clear and the effort exerted in the study appeared in it.

The discussion is well written.

The conclusion is very clear and significant 

Author Response

Dear Reviewer,

Thank you very much for your constructive criticism. We have edited our manuscript point by point according to your recommendations. In the following, your comments are printed in italics and the answers and additions to the manuscript are printed in blue. We hope that our manuscript is now suitable for publication in "Healthcare".

Thanks for giving me the chance to review this manuscript. I really enjoyed its reading .

The introduction is clear and to the point.

For the method, did the authors conduct any validation for the survey?

We did not validate the surveys ourselves. We used the validated questionnaires from the EORTC. We included in the M&M section:

“Both questionnaires have been validated by the EORTC.”

Was the study registered in ClinicalTrials.gov or approved by ethics'committee? The authors should write the registration number or the number of ethical approval indicating the ethics'committee

The study was not registered in ClinicalTrials. We have an ethical approval by the Ethics Committee of our University Hospital Erlangen. It is stated in the “Institutional Review Board Statement:” after the Conclusion:

"The study was conducted in accordance with the Declaration of Helsinki, and approved by the Ethics Committee of the University Hospital Erlangen (approval number: 3745 and date of approval: 17. April 2008)."

Did the patients sign the informed consent to be included in the study?

Yes, all patients were informed about the study and asked to participate. If they were willing, they signed an agreement to participate in the study.

We added in the M&M section:

“Written informed consent was obtained from all patients at the "front door", allowing for their participation in the study and the collection of their clinical data.”

The results are very clear and the effort exerted in the study appeared in it.

The discussion is well written.

The conclusion is very clear and significant 

Reviewer 2 Report

Dear authors

I read your manuscript "Impact of COVID-19 on Quality of Life in Long-term Advanced 2 Rectal Cancer Survivors" with great interest.  It is critical to investigate the quality of life of cancer patients.  The research is strong because it offers data for a 12-year period of follow-up on these individuals.  The data analysis was quite systematic, and the results were presented very clearly, however, some figures need to be revised. 

I would like the authors to consider the following comments:

Cancer classification should be mentioned in the introduction because it is a key aspect in determining these patients' QoL.

How reliable are the questionnaires used in this study? Please display the questionnaire's reliability results.  

The descriptive Table 1 presentation is extremely confusing.  Authors should simplify it so that it is readable.

Figure 2 (two-year follow-up figures) must be corrected for symptoms scores (B) for nausea and vomiting, as well as weight loss boxplots. The colors of the bars are missing.  This statistic is consistent with the previous years of follow-up.

The manuscript needs English revision.  There are certain grammatical errors: For example, in line 364, "therefor remains"

Finally, What does this study suggest or imply for clinicians who treat rectal patients? What are the study's implications for rectal patients?  Is this study having any policy implications?

The MS needs English editing. 

Author Response

Dear Reviewer,

Thank you very much for your constructive criticism. We have edited our manuscript point by point according to your recommendations. In the following, your comments are printed in italics and the answers and additions to the manuscript are printed in blue. We hope that our manuscript is now suitable for publication in "Healthcare".

Dear authors

I read your manuscript "Impact of COVID-19 on Quality of Life in Long-term Advanced 2 Rectal Cancer Survivors" with great interest.  It is critical to investigate the quality of life of cancer patients.  The research is strong because it offers data for a 12-year period of follow-up on these individuals.  The data analysis was quite systematic, and the results were presented very clearly, however, some figures need to be revised. 

I would like the authors to consider the following comments:

Cancer classification should be mentioned in the introduction because it is a key aspect in determining these patients' QoL.

We have changed one sentence in the introduction to mention the key aspect of cancer classification:

“Therefore, potentially influencing QOL-parameters, such as cancer classification, should be identified and analysed to understand their effects on patient outcomes.”

How reliable are the questionnaires used in this study? Please display the questionnaire's reliability results.

We did not test the reliability of the questionnaires ourselves. The QLQ-C30 is probably one of the most widely used questionnaires for quality of life and the QLQ-CR38 for quality of life in colorectal cancer. The validity and reliability of the questionnaires have been extensively tested by EORTC. We assume that the same will be true for our study. We therefore refer to studies on reliability from other studies.

Reliability scores have been published in other studies. The reliability results of the questionnaires can be found in the references [7] (doi:10.1093/jnci/85.5.365; QLQ-C30) and [8] (doi:10.1016/j.jclinepi.2014.09.021; QLQ-CR38).

The descriptive Table 1 presentation is extremely confusing.  Authors should simplify it so that it is readable.

To make the table easier to read, we have colored the different rows in grayscale, shortened the headings, avoided using percentage signs more than once, and rounded the numbers up.

Figure 2 (two-year follow-up figures) must be corrected for symptoms scores (B) for nausea and vomiting, as well as weight loss boxplots. The colors of the bars are missing.  This statistic is consistent with the previous years of follow-up.

No nausea and vomiting as well as no weight loss were reported by more than 75% of patients at certain times, such as at two years (Figure 2). Therefore, there are no boxplot bars to show (they go from 0% to 0%).

We have added the explanation in the caption of figures 1-4:

“If no boxplots are displayed, more than 75 % of the values are at 0 %.“

The manuscript needs English revision.  There are certain grammatical errors: For example, in line 364, "therefor remains"

The manuscript was reread and revised by us. The sentence you mention has been changed:

“Therefore, the additional significance of the rectal cancer diagnosis remains unclear.”

Finally, What does this study suggest or imply for clinicians who treat rectal patients? What are the study's implications for rectal patients?  Is this study having any policy implications?

The pandemic was an absolutely exceptional situation and hopefully there will not be another similar event so soon. That's why this work is important in terms of coming to terms with the pandemic. But it also shows that crisis situations have a significant impact on quality of life and that patients need all the more support here. This can certainly also be applied to other crisis situations. Our first analyses of quality of life since the beginning of the war in Ukraine indicate that quality of life is again affected, although patients are much less directly affected by the war than by the pandemic.

It is very difficult to assess what it is about the pandemic that has led to the quality of life reductions. Whether it was the threat of the pandemic or the restrictions that caused it. Of course, different or fewer restrictions might have caused other worries and fears. Thus, we cannot give advice to policy makers.

We added to the discussion:

“This study focused on the impact of the pandemic on rectal cancer patients. The immediate lesson would be that during a pandemic, there is a need to be more responsive to patients and their concerns and fears, as the crisis has an additional negative impact on QOL on top of the catastrophe of the disease itself. However, what this work certainly shows is that patients' QOL is affected by a crisis. And probably other crises, such as economic, political, or environmental crises, also affect QOL. Patients should be offered more psychological support during such periods.

What we cannot assess is whether the pandemic itself, through the threat of COVID-19, or the restrictions imposed by policymakers, or both, caused the decline in QOL. We are even less able to assess whether fewer restrictions would have helped to improve QOL."

Reviewer 3 Report

This is an interesting article by Blasko et al. that evaluates the impact of COVID19 on QoL of long-term advanced rectal cancer survivors. The manuscript is well written and the results are adequately presented. The introduction  provides a concise summary of the current evidence, whereas the methods section delineates the applied methodology. The provided figures and tables are of high quality and quite informative. A few comments though:

- Please consider providing further information regarding patient screening (consecutive or selected patients?, retrospective or prospective database?)

-Please provide further base demographic data of the eligible patients (BMI,ASA, age, operation type)

- The study subgroups should be matched on the basis of these confounders to further reduce bias. Matched cohorts should be then reanalyzed.

-Results section should be significantly minimized to increase readability.

-Please provide a strengths and limitations section in the Discussion

Overall the manuscript is well written. Minor linguistic revisions are required

Author Response

Dear Reviewer,

Thank you very much for your constructive criticism. We have edited our manuscript point by point according to your recommendations. In the following, your comments are printed in italics and the answers and additions to the manuscript are printed in blue. We hope that our manuscript is now suitable for publication in "Healthcare".

This is an interesting article by Blasko et al. that evaluates the impact of COVID19 on QoL of long-term advanced rectal cancer survivors. The manuscript is well written and the results are adequately presented. The introduction  provides a concise summary of the current evidence, whereas the methods section delineates the applied methodology. The provided figures and tables are of high quality and quite informative. A few comments though:

- Please consider providing further information regarding patient screening (consecutive or selected patients?, retrospective or prospective database?)#

All patients treated for advanced rectal cancer in our department were asked if they would like to participate in the study. A total of 594 patients were contacted, of whom 489 agreed to participate in the study and completed at least one questionnaire. Patients were consecutively interviewed and data were collected prospectively. Clinical data were collected retrospectively from patient records.

We added to the M&M section:

“All patients treated for advanced rectal cancer in our department were consecutively invited for participation in the study. A total of 594 patients were asked to participate in the study, and 489 patients completed at least one questionnaire. Initially, the Non-COVID-19 study group consisted of 328 patients and the COVID-19 group consisted of 61 patients. The discrepancy between the number of patients who completed the first questionnaire and the patient sample size can be explained by the fact that some patients chose to complete the second questionnaire but not the first and vice versa. On average, each patient completed 4.27 questionnaires. Consequently, patients were interviewed consecutively and data were collected prospectively. Clinical data were collected retrospectively from patient records.”

-Please provide further base demographic data of the eligible patients (BMI,ASA, age, operation type)

We have very incomplete BMI data because of the retrospective analysis of the data, so it makes no sense to list this subgroup. The ASA data are not available because they are only available in non-digital form and thus could not be viewed by us. Patients ranged in age from 15 to 93 years at the start of therapy (day 0), with a mean age of 63.5 years. The age of the Non-COVID-19 cohort was 63.5 years, range 23-86, the varying Non-COVID-19/COVID-19 was 63.1 years, range 15-90, and the COVID-19 was 64.3 years, range 41-93 years.

We added in the M&M section:

"The study includes 489 patients from the University Hospital Erlangen, Germany, with advanced rectal cancer who were treated with neoadjuvant radiochemotherapy followed by surgical excision of the entire mesorectum, corresponding to a total mesorectal excision (TME).

“Patients ranged in age from 15 to 93 years at the start of therapy (day 0), with a mean age of 63.5 years."

A row with the age of the patients of the respective subgroups was added to the descriptive table 1.

- The study subgroups should be matched on the basis of these confounders to further reduce bias. Matched cohorts should be then reanalyzed.

Unfortunately, we do not have the BMI and ASA values, so we cannot do an analysis for them. The age is very similar in the three groups and there is no reason to expect a bias in the results. All patients underwent the same surgery.

-Results section should be significantly minimized to increase readability.

We agree that the Results section is quite long. However, we do want to describe the largest/most interesting score differences of each of the ten figures in this section. Also, the exact percentages of the symptom/functional scores are not shown in the figures themselves, which supports this. Therefore, we feel that a long results section is unavoidable.

-Please provide a strengths and limitations section in the Discussion

The limitations of our study can be found in the Discussion in paragraphs two and three. A strength section has been added:

“Strengths of our study include its prospective nature, in which changes in functioning and symptom perception were frequently assessed during the different phases of therapy. Therefore, changes could be accurately measured. In addition, the large number of patients over more than a decade resulted in a great patient sample and thus a large amount of study data. Questionnaires several years after therapy represent a long follow-up period and support this on an individual basis.”

Reviewer 4 Report

Dear Editor,

The article entitled “Impact of COVID-19 on Quality of Life in Long-term Advanced Rectal Cancer Survivors” has important scientific message and generally well written; A few issues, however, need to be addressed:

Please carefully proof-read spell check to eliminate grammatical errors.

Age; sex and stage are not necessary as keywords.

Minor editing of English language required.

Author Response

Dear Reviewer,

Thank you very much for your constructive criticism. We have edited our manuscript point by point according to your recommendations. In the following, your comments are printed in italics and the answers and additions to the manuscript are printed in blue. We hope that our manuscript is now suitable for publication in "Healthcare".

The article entitled “Impact of COVID-19 on Quality of Life in Long-term Advanced Rectal Cancer Survivors” has important scientific message and generally well written; A few issues, however, need to be addressed:

Please carefully proof-read spell check to eliminate grammatical errors.

The manuscript has been reread and carefully revised by us.

Age; sex and stage are not necessary as keywords.

The keywords age, sex and stage have been removed.

Reviewer 5 Report

Dear authors,

I have carefully read your article, although from the beginning I must note that it is quite difficult to follow in certain parts. I think that revising the English language with the use of shorter sentences would be desirable.

I think the study methodology is biased or was not well explained. The first problem is related to the number of patients who were included in the study. Are there 489 patients or only those who answered the questionnaires?

If they are only "before COVID-19 (87 to 328 patients) and during COVID-19 (28 to 63 patients)", then from my point of view only these are the groups studied.

It is not clear to me if patients were included along the way or what happened to the patients who died? I recommend that the inclusion criteria and the exclusion criteria be clearer.

In the presented methodology, there is no reference to how many of the studied patients presented with COVID19? Viral infection may influence the results of the study. It is known that this infection presented numerous symptoms and signs and some were evaluated through questionnaires.

If patients were included during the study, they all answered the eight questionnaires. If not, what was compared?

Based on the above, I cannot comment on the results obtained.

I want to note that I don't think the conclusion related to weight loss in T4 patients compared to the rest of the stages is relevant. The presence of cachexia in the metastatic stage is very well known.

I agree that the study is very vast, but the data analysis must be improved.

Major revision

Author Response

Dear Reviewer,

Thank you very much for your constructive criticism. We have edited our manuscript point by point according to your recommendations. In the following, your comments are printed in italics and the answers and additions to the manuscript are printed in blue. We hope that our manuscript is now suitable for publication in "Healthcare".

Dear authors,

I have carefully read your article, although from the beginning I must note that it is quite difficult to follow in certain parts. I think that revising the English language with the use of shorter sentences would be desirable.

I think the study methodology is biased or was not well explained. The first problem is related to the number of patients who were included in the study. Are there 489 patients or only those who answered the questionnaires?

All patients treated for advanced rectal cancer in our department were asked if they would like to participate in the study. A total of 594 patients were contacted, of whom 489 agreed to participate in the study and completed at least one questionnaire. On average, each patient completed 4.27 questionnaires. Consequently, patients were interviewed consecutively and data were collected prospectively. Clinical data were collected retrospectively from patient records.

We added to the M&M section:

“All patients treated for advanced rectal cancer in our department were consecutively invited for participation in the study. A total of 594 patients were asked to participate in the study, and 489 patients completed at least one questionnaire. Initially, the Non-COVID-19 study group consisted of 328 patients and the COVID-19 group consisted of 61 patients. The discrepancy between the number of patients who completed the first questionnaire and the patient sample size can be explained by the fact that some patients chose to complete the second questionnaire but not the first and vice versa. On average, each patient completed 4.27 questionnaires. Consequently, patients were interviewed consecutively and data were collected prospectively. Clinical data were collected retrospectively from patient records.”

If they are only "before COVID-19 (87 to 328 patients) and during COVID-19 (28 to 63 patients)", then from my point of view only these are the groups studied.

Yes, these are the groups studied. The first questionnaire was answered by 328 patients in the Non-COVID-19 study group and by 61 patients in the COVID-19 study group. The discrepancy between the number of patients who completed the first questionnaire and the patient sample size can be explained by the fact that some patients chose to complete the second questionnaire but not the first and vice versa.

We added to the M&M:

“A total of 594 patients were asked to participate in the study, and 489 patients completed at least one questionnaire. Initially, the Non-COVID-19 study group consisted of 328 patients and the COVID-19 group consisted of 61 patients. The discrepancy between the number of patients who completed the first questionnaire and the patient sample size can be explained by the fact that some patients chose to complete the second questionnaire but not the first and vice versa. On average, each patient completed 4.27 questionnaires.”

It is not clear to me if patients were included along the way or what happened to the patients who died? I recommend that the inclusion criteria and the exclusion criteria be clearer.

Death was not used as an exclusion criterion. Consent to participate and completion of at least one questionnaire was required. However, an average of 4.27 questionnaires were filled out by each patient.

In the presented methodology, there is no reference to how many of the studied patients presented with COVID19? Viral infection may influence the results of the study. It is known that this infection presented numerous symptoms and signs and some were evaluated through questionnaires.

During the six weeks of treatment, no cases of COVID-19 occurred in our patients. We do not know if any patients became ill during the follow-up period.

We added to the limitations section:

“Viral infections of COVID-19 may also cause symptoms similar to some of those listed in the questionnaire and thus affect the results. During the six weeks of radiochemotherapy, there were no cases of COVID-19. However, we do not have these data for further follow-up and cannot exclude bias due to possible COVID-19 disease.”

If patients were included during the study, they all answered the eight questionnaires. If not, what was compared?

No, on average they answered 4.27 questionnaires. Of course, patients are free to answer the questionnaires, and so there are patients who decided not to participate in the survey after completing some questionnaires. Patients, who started their therapy after the start of COVID-19 could not have completed the four-year follow-up, as four years haven’t passed since the start of the pandemic. Therefore, individual questionnaires rather than patients were compared to each other, depending on when the questionnaires were filled out (before or during COVID-19).

In the third paragraph of the Discussion, we hint on that: “Since voluntary questionnaires can be declined by patients at any time, there is a selection bias in the study group.” and in the same paragraph “Excessively anxious patients who do not participate would produce biased and embellished results.”

This is also the reason why a Varying Non-COVID-19/COVID-19 group was created for the descriptive “Table 1”, as some patients changed their group over time, as explained in the M&M section: “Each patient was assigned to the “Non-COVID-19”, “COVID-19” or for reasons of clarity specifically created “Varying Non-COVID-19/COVID-19” column. The latter includes all patients who changed groups over time and completed surveys before and during COVID-19.”.

However, an average of 4.27 questionnaires were filled out by each patient.

Based on the above, I cannot comment on the results obtained.

I want to note that I don't think the conclusion related to weight loss in T4 patients compared to the rest of the stages is relevant. The presence of cachexia in the metastatic stage is very well known.

We have edited this part of the Discussion to make it clearer:

“Compared to the time before COVID-19, cT4-staged rectal cancer patients lost more weight during the pandemic than their cT1-3 counterparts, which is directly linked to higher appetite loss. COVID-19 infections and worries could have contributed to this. Delays in seeking medical care may have led to more frequent proliferation of cancer cells. Because of their exponential growth, this would have a more consuming effect on patients with higher stages of cancer, explaining the findings.”

The weight loss of cT4-staged rectal cancer patients is compared before and after the start of the COVID-19 pandemic. The same is then done for cT1-3-staged rectal cancer patients. These differences are then compared to each other and show a large gap that cannot be explained by cachexia. Figure 10W illustrates this.

I agree that the study is very vast, but the data analysis must be improved.

Round 2

Reviewer 3 Report

The authors managed to answer to most topics raised by the reviewers. However some major issues were not addressed. More specifically:

- Patient characteristics should be reported

-Group matching is advisable to eliminate the possibility of bias

- Results section should be drastically revised.

Author Response

Dear Reviewer,

Thank you very much for your constructive criticism. We have edited our manuscript point by point according to your recommendations. In the following, your comments are printed in italics and the answers and additions to the manuscript are printed in blue. We hope that our manuscript is now suitable for publication in "Healthcare".

The authors managed to answer to most topics raised by the reviewers. However some major issues were not addressed. More specifically:

- Patient characteristics should be reported

A patient screening flowchart has been added with a table that distinguishes between patients who completed subsequent questionnaires without completing the first initial questionnaire and patients who completed both the first and subsequent questionnaires, showing that more than 80% of respondents always completed the subsequent questionnaires and the first questionnaire.

Unfortunately, we do not have the BMI and ASA values.

-Group matching is advisable to eliminate the possibility of bias

We believe that the new Figure 1 shows that the majority of patients completed the questionnaires regularly and only a small proportion < 20% did not respond regularly. Table 1 clearly shows that the patient groups were almost identical in terms of age, sex and tumor stage and there is no evidence of bias. Although we do not have reliable data on BMI or ASA, with such a large number of patients and the consecutive enrolment of patients through this randomization, there should be no significant differences between patients.

- Results section should be drastically revised.

Parts of the Results section have been moved to the Materials and Methods section (as shown below), so the Results section has been shortened:

“As part of the patient sample used in this paper with data extending to June 2021 was examined by another study for the time points one to four, we focused on the annual follow-ups, developments throughout all questionnaires - including by sex and age - and the QOL dependent on the clinical T-stage of the cancer. To illustrate the development across all questionnaires, the combined means of all nine functional scores and the combined means of all fourteen symptom scores were compared before and during COVID-19 for all eight questionnaires. Further analyses were performed across all eight time points for subgroups of sex, comparing males to females, and age, com-paring patients 65 years and older to patients younger than 65 years. In addition, sub-groups were created based on the clinical T stage of the patients’ rectal cancer and plotted over the course of 70 days, including four questionnaires. A subgroup with less advanced cancer (cT1 – cT3) was compared to a subgroup with more advanced cancer (cT4). Since the less advanced cancer subgroup includes mostly cT3 stage cancers, less advanced has to be seen in relative terms.”

Reviewer 5 Report

the authors made changes in the article, but I think that the interest for readers remains average. It is not clear to me how the pandemic influenced these results or if they were influenced by it.

I do not think that this article should be published in this form.

Minor editing is need it.

Author Response

the authors made changes in the article, but I think that the interest for readers remains average. It is not clear to me how the pandemic influenced these results or if they were influenced by it.

I do not think that this article should be published in this form.

Thank you for your feedback and for taking the time to review our work. While we appreciate your perspective, we respectfully disagree with your views on the article's interest to readers. We believe that we have shown an impact of the pandemic on the quality of life of rectal cancer patients in the form of a trend of decreasing scores. Additionally, a few other changes have been made: adding a patient screening flowchart and moving some sections from the Results section to the Materials and Methods section.